# Perceived Overqualification and Job Crafting: The Curvilinear Moderation of Career Adaptability

**Hyung Rok Woo** 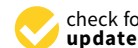

Department of Business Administration, Mokpo National University, Mokpo-si 58554, Korea; hrwoo@mokpo.ac.kr

**Abstract:** Developments in direct and indirect educational systems have increased the number of individuals with competencies that are higher than the required expectations of their current jobs. This concept of overqualification is drawing interest today, while underqualification was the focus in the past. Currently, research on perceived overqualification (POQ) has focused on its negative influences; however, this study aimed to explore the positive and nonlinear influence of POQ on job crafting and the moderating effect of career adaptability on these functions. Data were collected from 257 individuals in three Korean telecom companies. The results of hierarchical regression analysis indicate that POQ had a reverse U-shaped nonlinear influence on job crafting, indicating that an appropriate level of POQ can drive job crafting, leading to performance improvements in organizations and individuals. Moreover, career adaptability moderated the relationship between POQ and job crafting. When career adaptability was at an average or high level, the reverse U-shaped nonlinear influence of POQ on job crafting strengthened. These results are expected to assist in creating both an environment to reduce the negative influence of POQ and healthy sustainability in human resources development.

**Keywords:** perceived overqualification; career adaptability; job crafting; moderated U-shaped relationship; sustainable career development

## 1. Introduction

The crash and uncertainty of the global economy has led to a reduction in employment opportunities, and in turn has led to lower employment quality. While employment naturally reduces the risks associated with unemployment, there may be other, psychological, costs associated with unsatisfactory treatment and tasks at work [1,2]. Underemployment particularly, overqualification, has therefore become a significant social issue especially in developed countries.

Overqualification refers to situations where an individual has competencies that exceed the levels of knowledge, education, and experience required to perform the tasks associated with their job [3,4]. Given the significant increase in academic inflation, the phenomenon of overqualification is common not only in Korea but also in countries such as the United States and China and European countries [5]. According to studies from the United States, Canada, and the United Kingdom, 1 in 3 employees are considered to be overqualified [6,7]. Furthermore, the issue of disguised employment, i.e., working in a position that is of lesser quality than one's previously held position, is also rising due to reduced employment opportunities.

Organizational effectiveness is reduced by the individual's negative self-perception, best described as "the big fish in a small pond" [8,9]. Besides actual overqualification, the perception of overqualification and the level of overqualification perceived by the individual also influence the behavior and mental health of employees significantly [10,11]. Perceived overqualification (POQ) refers to the degree that individuals feel overqualified on account of the comparison between the employees' abilities and the

tasks assigned to them at the workplace [12]. The feelings are reported to have a negative influence on the performance of individuals and therefore organizations. POQ leads to boredom and reduces work engagement [13], undermining desirable attitudes and work satisfaction and can ultimately cause individuals to leave the organization [12,14]. Furthermore, POQ has a negative influence on work performance, team performance, organizational citizenship behavior, organizational engagement, and creative performance, leading to undesirable results [15–18], including higher turnover intention and counterproductive behavior [19,20]. The overqualification focused on its negative aspects seemed to be the impetus behind the negative performances in organization.

The question, however, remains unanswered: Do employees who perceive themselves as being overqualified always experience lower job satisfaction and higher turnover intention, and in turn, does this negatively influence the organization? This study has focused on a small number of claims that employees who perceive themselves as being overqualified do not always negatively influence the organization and that utilizing their potential can have a positive influence on the organization, since such employees are more likely to be more capable and experienced compared to regular employees. There exists, however, limited empirical research [21] on how employees healthily resolve and overcome POQ, and the mechanisms used in the process.

As such, this study presumes job crafting as a behavior that reduces the negative influence of POQ and aims to shed light on this relationship. As overqualification is based on an imbalance between individual competencies and job requirements, one direct solution is job redesign aiming for person–job fit. In relation to job redesign, job crafting, which involves the proactive redesigning and reinterpreting of one's job [22], is a pathway to overcome POQ. Therefore, the conceptual characteristics of job redesign, involves bestowing new meaning on one's job and voluntarily reconstructing the methods and boundaries of one's work. Thus, job crafting can be regarded as the result of healthy improvement activities that help individuals overcome overqualification [23,24].

Furthermore, this study verified career adaptability as a factor that moderates the relationship between POQ and job crafting. Career adaptability, based on the career construction theory, is the mental strength of the individual to effectively overcome challenging situations that could not have been predicted in the career development process [25]. Depending on the level of career adaptability, POQ can either lead to frustration or to opportunities for a challenge. Therefore, career adaptability is an individual trait that acts as an important situational factor when the POQ-experiencing individual attempts job crafting.

Therefore, the purpose of this study is to uncover the methods used to reduce the negative influences of POQ and reveal the dynamic relationship between POQ, career adaptability, and job crafting. Furthermore, it is expected that this study will provide important implications related to economic benefits such as reducing underemployment, improving employment quality, and driving successful business activities.

## 2. Theoretical Background

### 2.1. Perceived Overqualification

Overqualification refers to a state wherein employees have higher levels of competencies, such as skills, knowledge, abilities, education, and experience, than what is required to perform their jobs [4,12]. Overqualification is a special type of underemployment, which refers to the state of being employed but in an improper manner [26,27]. Along with overqualification, underemployment also includes situations in which individuals are paid less than what they were paid before, as well as having unstable employment, such as holding a temporary position [28]. Among these concepts, overqualification is considered to be one of the indicators that generally represent and are related to other types of underemployment [29].

While it is difficult to accurately measure overqualification, objective levels of overqualification can, to some extent, be measured by the gap between job requirements and individual competencies.

The subjective perception of overqualification felt by the individual, their experience of it, and its level of perceived severity is significant [8,30]. This is because subjective perception and psychological assessments influence the individual's actual work behavior [31,32]. The majority of research in organizational behavior utilize perceived overqualification (POQ), a subjective measure of overqualification as their focus, on account of it being better able to predict the attitudes and behaviors of employees [33,34].

Ultimately, POQ refers to the individuals' belief that they possess higher levels of skills, experience, and knowledge than is required by their job. Many researchers have established the theoretical composition and definition of POQ and have developed scales to measure it. Johnson, et al. [35] developed a POQ scale composed of a "mismatched facet," forming a picture of an employed individual's belief that they have competencies beyond what is required by their job, and a "no-growth facet," which refers to the perception that the individual is offered limited opportunities to obtain and utilize new job-related skills. Furthermore, Maynard, et al. [36] asserted that the mismatched facets were only reflective of the basic nature of POQ, which was "having more education, training, and experience than what is required for the job." As such, research [3,37] into POQ sometimes employs the scale developed by Johnson, Morrow, and Johnson [35], but the majority of research focuses solely on mismatched facets, omitting no-growth facets.

Typically, POQ is known to have a negative influence on job attitude and employee behavior. The person–job fit theory explains overqualification as the person–job misfit in which one's qualifications do not match job demands; in turn, POQ causes negative individual and organizational performances [38]. Higher POQ is associated with feelings of helplessness among employees, as they are not able to sufficiently utilize their knowledge and experience [14]. The relative deprivation theory [39] explains the psychological nature of overqualification better than the person–job fit theory [12,40]. Employees could feel relative deprivation in failing to meet their desires and expectations for their job [31,41]. Because of this relative deprivation, overqualification is often accompanied by unfair treatment, such as being excluded from opportunities of education and self-development [41]. Such experiences reduce the individual's desire for growth and result in a negative influence on job performance [3], team performance [42], and organization citizenship behavior [43]. This in turn leads to higher job boredom [44], turnover intention [45], withdrawal behaviors [14], and counterproductive work behavior [38].

A small number of studies, however, insist on the positive influence of POQ on employee performance. There are some claims that employees who feel overqualified do not always negatively impact on their organization but instead could actually affect the organization positively. POQ could have both positive and negative influences on their work performance [41]. According to Zhang, Law, and Lin [9], individuals with POQ may experience boredom associated with their tasks; however, they easily achieve job objectives, with a high degree of self-efficacy and perform beyond what is minimally required by the job. With this in mind, it is clear that research incorporating various perspectives on POQ, such as those focusing on its positive aspects is necessary to more fully understand the phenomenon of overqualification. van Dijk, Shantz, and Alfes [21] argue that POQ, as positive performance is achieved, has potential advantages from human capital and social learning perspectives. There has been empirical evidence of adequate qualification and POQ of colleagues in the same group, with the situational condition of individual POQ having a positive influence on in-role and extra-role performance [8]. Luksyte and Spitzmueller [46] showed empirically that employees could improve their creative performance when they experienced developmental idiosyncratic deals with supervisors as well as perceived organizational support. Some research indicates that higher POQ leads to better job performance [47,48]; Erdogan and Bauer [3] provided empirical evidence for the positive relationship between sales performance, as a type of job performance, and POQ.

As such, there are mixed results on the positive and negative influences of POQ, which indicates that further work is required to find consistent results [8,49]. This study focused on the assertions of

Russell et al. [50], who claimed that employees with high POQ could achieve positive job performance when they are provided with career development opportunities such as job crafting.

### 2.2. Job Crafting

In a stable industrial environment, it is important to understand and adhere to roles and job scopes defined in the job description. In a corporate environment on the other hand, with higher uncertainty, behaviors including voluntary formulation of job boundaries and building relationships with stakeholders becomes more important [23,51,52].

Job crafting is defined as the physical and cognitive behavior employees used to initiate job-relevant changes in their work and relationships [53], but it is differentiated from traditional job redesign in that it is not driven by the human resource department through official processes and methods [54]. Job crafting can be performed by the individual without official approvals [55,56]. In other words, it refers to proactive and voluntary efforts by the employee to seek change in their own work [22,57]. This proactive attitude is in contrast with the reactive attitude of performing their job according to the guidelines. Wrzesniewski and Dutton [53], who first proposed the concept of job crafting, explain that it is based on job redesign at a superficial level, but it includes proactive changes initiated by individuals to improve characteristics of their work and change their perceptions about what their work means to them, as opposed to a top–down approach.

Wrzesniewski and Dutton [53] divided job crafting into task crafting, relational crafting, and cognitive crafting. Task crafting refers to the activities associated with improving the processes, methods, and formats of tasks by altering the number or boundaries of tasks. Relational crafting refers to changing the level of interactions with people relevant to carrying out the job, such as the scope, frequency, and the degree of meetings with such people. Cognitive crafting refers to activities that involve reinterpreting, bestowing meaning, and redefining one's job, so as to result in perceptions of higher value.

There are three main drivers for individuals to engage in job crafting [53]. First is the desire for control. This desire emerges from the need to avoid negative situations, such as being excluded from jobs, and from lacking the ability to assert agency over the processes and results of jobs. Second is the desire to build a positive image for oneself. To build a positive image, individuals must strive to reconstruct meaning and identity for their work, so that in carrying it out they are seen as being honorable and valuable. Third is the desire to form social relationships. To form good social relationships with those around them, individuals must strive to complete tasks that are given to them, while attempting to bring change to their jobs or to engage in new activities.

In contrast, European research on job crafting is based on the job demands–resources model [58], which categorized the traits of a job into demands and resources. Job demands refer to the cognitive and emotional effort required of the individual in situations related to their job. This includes factors relating to job difficulty and psychological costs, such as task intensity, task complexity, time pressure, and role ambiguity. In contrast, job resources refer to various resources and support that the individual can draw on to complete their job. These resources include factors from the job characteristics model [59], such as feedback and autonomy, as well as social support from their peers and superiors. When there is a balance between job demands and job resources, employees are able to conduct their jobs smoothly and effectively; naturally however, this does not hold true in the case of an imbalance.

In other words, job crafting is regarded as the behaviors that reduce the imbalance between job demands and job resources. Job crafting is the proactive behavior of seeking a balance between finding resources required by one's jobs and taking on challenging tasks, as well as reducing the physical and psychological costs required by one's job [60,61]. Tims, Bakker, and Derks [22] presented four specific directions of job crafting. Individuals either decrease hindering job demands or increase challenging job demands. They can also increase structural job resources, such as task diversity, autonomy, and growth opportunities, or increase social job resources such as coaching and feedback.

From the perspective of the three desires of engaging in job crafting, as well as behaviors to reduce imbalance between job demands and job resources, the majority of employees who perceive overqualification in themselves are known to first consider or attempt innovative behavior or job crafting to improve the suitability of the contents of their work [62–64]. Those who craft their jobs optimize their person–job fit over time [44,56]. Thus, job crafting could be driven by awareness regarding a poor person–job fit, such as overqualification. Employees who perceive overqualified would engage in job crafting to improve their POQ, and if successfully implemented, may experience positive results such as job commitment. Several researchers e.g., [49,50] have already started to argue that employees with POQ tend to positively craft their job.

The majority of research on POQ, however, has focused on its negative influences on job attitude and behavior of the employees in an organization. It is worth noting their claim that overqualification is a stressful experience for employees [46]. That means that the effects of stress are not universally negative. It is commonly known that negative factors such as stress have a curvilinear relation with various performance indicators. In a similar vein, we propose the following research hypothesis involving the inverted U-shaped relation between POQ and job crafting.

**Hypothesis H1.** *POQ will have an inverted U-shaped relation with job crafting.*

In other words, we try to carefully examine the different effects depending on the level of overqualification. When POQ is at a low-to-moderate level, employees can engage in additional activities, such as job crafting [63]. Overqualified workers are likely to have additional time because they can easily or rapidly finish delegated tasks without utilizing all their capabilities [65]. A goal of job crafting is to make better perceptions regarding person–job fit [53]. Overqualified workers would go on with job crafting if POQ is low enough to believe that they can improve it with their surplus capabilities. As the lack of person–job fit becomes deeper, they would realize that job crafting alone could not improve it. The proactive job crafting would eventually diminish if POQ is too high, causing boredom and dissatisfaction. To sum up, individuals engage in job crafting at a manageable level of POQ; however, extreme levels of POQ will harm job crafting. To uncover mixed results on the positive and negative effects of POQ, it is important to consider the inflection point in the relationship between POQ and job crafting.

## 2.3. Career Adaptability

To answer the question of how individual traits can lead to positive POQ, this study attempts to identify career adaptability as a moderating factor. There are claims that studies on overqualification to date have focused on individual overqualification without the consideration of the individual's surrounding environment and context. Many researchers have asserted that the extent to which overqualification is found in individuals is impacted by the surrounding environment [66], and thus it is important to consider any contextual effects of the employees' environment and avoid emphasizing the overqualification of individuals alone [67].

Today, employee career development is becoming nonlinear, boundaryless, and diverse in nature [68], meaning workers are now required to respond appropriately to their job environments and possess active attitudes that equip them to control their own environment. Given such changes in the work environment, career adaptability is seen as the desire to improve the individual–environment adaptability in the field of career development and is regarded as a factor that contributes to building a successful career [69]. As such, there is an increasing body of research on the relationship between POQ and variables relating to career development [11,70,71].

Initially, the term "career adaptability," born out of career maturity, was first proposed by Super and Knasel [72]. It is a concept that includes the ability or willingness of the individual to proactively change one's attitudes, competencies, and behaviors to adapt to the environment [25,73]. Until recently, it was used in a mixed manner with other concepts such as career planning and career

exploration [74], career identity [75], protean or boundaryless career [76], and career preparation [77]. Savickas [25,78] applied social constructionism to the theory of career development, which focused on individual–environment adaptability, merging existing assertions and creating a new basis for career adaptability research [79]. He believed that a career, as a concept, is not to unfold something; rather, it is to construct self and identity along with the environment in which work is performed.

Savickas [25,73] delineated career adaptability into the following four categories. First, career concern, which involves the planning of careers, is a process that involves constructing a blueprint of one's career path or a career vision, preparing for specific activities and paving the way to achieve that career vision. This is a concept that differs from mere interest; rather, it involves reflection on the individual's present and past and connects their findings with career-related objectives and aspirations.

Second, career control is the perception of the individual as the person in charge, or the decision-maker of their own future. Career control is related to concepts such as locus of control, autonomy, and proactiveness and refers to the ability to prioritize and determine career alternatives.

Third, career curiosity refers to exploring information for career development. The subjects of such an exploration are the individual themselves and the environment. Self-exploration targets individual desires, interests, values, and objectives, as well as the knowledge and skills required for future career objectives. Furthermore, self-exploration is closely related to the construction of one's career identity [75]. Environment exploration refers to the act of collecting information on industries, firms, jobs, and tasks to secure the availability of many alternatives when facing career decisions. Career curiosity and career interest, however, have contrasting traits. Career curiosity refers to open-ended thinking and a wide area of interests, whereas career interest involves engaging in specific objectives and has a comparatively narrower direction.

Fourth, career confidence is defined as the belief in one's success at overcoming challenging obstacles in the career development process, and refers to self-efficacy in succeeding in attaining career objectives [80]. A high level of career confidence instills competence and is displayed as the non-hesitant attitude of passionately responding to the requirements of the job and the job environment.

Finally, all individuals should craft their jobs throughout work life in order to optimize their person–job fit [57,64]. It is necessary to consider not only current job design but also long-term career development. We suggest an integrative theoretical framework that draws upon both job design view and career development view in order to delineate the nature of the relations among POQ, job crafting, and career adaptability. Career adaptability gives the employee a view of the future and helps to situate them in an environment that is evolving. This means that the current job and tasks should be shaped to fit the employee's conceptualization of their career, shaping the job to suit the career, as it were. This future-oriented perspective may lead one to see the current job as being a temporary step in a lifelong career, and thus, one is more able and willing to shape that job or task. Career adaptability is the mental strength enabling one to effectively process challenging situations that could not have been predicted in the career development process and has a positive influence on employability [67,81], career success [82,83], work volition [84], entrepreneurial orientation [85], and organizational commitment [86].

As discussed above, considering the concepts and influence of career adaptability, this study is able to predict its positive role even when individuals are faced with the issue of overqualification, which was not predicted before the job was assigned. Higher motivation to deal with POQ can lead to the positive effects of POQ; however, lower motivation can lead to POQ being viewed as destructive. We introduce career adaptability as one of the incentivizing factors. Yang, et al. [87], in a study of corporate human resource experts, asserted that higher levels of career adaptability can reduce overqualification as individuals can access opportunities such as delegating authority from managers.

Career adaptability is strongly associated with passion for the future and resilience and is the competency that allows for the maintenance of a positive psychological state [67,69,88]. In short, proactivity requires adaptivity [23,89]. As such, higher career adaptability can lead to POQ being

viewed as an incentivizing domain that can be challenged, whereas lower career adaptability can lead to the perception of POQ as a domain evoking an avoidance response [11,50]. Individuals with high career adaptability place positive values on risk factors, do not give up when faced with adversity, come up with new alternatives, and possess strong problem-solving tendencies [87]. This strengthens the positive effects of POQ on job crafting. However, those with lower career adaptability may comparatively lack those psychological resources. They may give up when faced with risk factors and difficulties. In other words, POQ may lead to an aversive and adverse response.

In summary, with higher career adaptability, POQ at appropriate levels can activate its positive functions, including achievement needs and challenges. In such situations, job crafting may be high despite high levels of POQ. However, lower levels of career adaptability may lead to individuals experiencing the adverse functions of POQ, leading to POQ having a negative influence on job crafting. Based on this logic, this study believes that career adaptability will have a moderating effect with an inverted U-shape.

**Hypothesis H2.** *Career adaptability will enhance the curvilinear relationship between POQ and job crafting; specifically, the inverted U-shaped relation is stronger for employees with high career adaptability and flatter for employees with low career adaptability.*

## 3. Materials and Methods

### 3.1. Research Subjects and Data Collection

The data collection for hypothesis verification was conducted in April 2019, within three Korean telecom companies: M, S, and P. The survey was conducted with all employees working in the call centers of the three companies, who had work experience of five years or less. Their responsibility is relatively simple and normalized—responding to inbound calls from the customers. Five years of work experience marks a period in which individuals may perceive overqualification prior to adapting to new employment and associated tasks and is therefore suitable for the current study. Participants were recruited publicly after receiving approval from the human resources and other relevant departments in the companies. The survey was delivered to the recruited individuals using the random assignment method. The data collection method used contact information in the sampling frame, and surveys were distributed and collected using personal e-mail addresses. A total of 265 responses were collected. Excluding incomplete responses, 257 responses were set as the effective sample and used in hypothesis verification.

### 3.2. Measurement Scales

Survey items were measured using a 6-point Likert scale ranging from 1 point ("strongly disagree") to 6 points ("strongly agree"). The 6-point scales are effective in simultaneously obtaining validity and reliability [90] and are likely to follow normal distributions [91]. Information on participants' gender, educational level, and their employers were processed as dummy variables and were applied as control variables along with their age.

First, POQ was measured using the nine items developed by Maynard, Joseph, and Maynard [36]. This scale assesses the level of perceived overqualification based on education, experience, knowledge, skills, and ability. A sample item was "I feel confident in designing new procedures for my work area." Job crafting was measured using the Job Crafting Questionnaire (JCQ) developed by Slemp and Vella-Brodrick [92]. The JCQ is composed of 15 questions, with five questions being allocated to each of the three sub-factors of job crafting asserted by Wrzesniewski and Dutton [53]; task crafting, relational crafting, and cognitive crafting. Career adaptability was measured using the Career Adapt-Ability Scale (CAAS) developed by Savickas and Porfeli [66]. The CAAS is composed of 24 items, with six questions assigned to each of the four sub-factors of concern, control, curiosity, and confidence. The analysis process was as follows. The reliability and validity of the measurement

scales were verified. Then, hierarchical multiple regression was used to confirm the nonlinear effect of POQ on job creation, as well as the moderating effect of career adaptability.

### 3.3. Reliability and Validity

Cronbach's α was calculated to check whether internal consistency was present in the measurement items of each of the variables. To verify construct validity, confirmatory factor analysis was conducted (see Table 1). The Cronbach's α of all major variables used in this study exceeded 0.70. The results of the confirmatory factor analysis indicated that the standardized regression weights of all factors were higher than 0.50, indicating that construct validity was secured. Composite reliability (CR) ranged higher than 0.7, between 0.811 and 0.889. Average variance extracted (AVE) ranged between 0.596 and 0.661, which was higher than the general criteria of 0.50, thus confirming convergent validity.

**Table 1.** Results of internal consistency and confirmatory factor analysis.

| Constructs | Cronbach's α | AVE | CR |
|---|---|---|---|
| POQ | 0.873 | 0.661 | 0.811 |
| Career adaptability | 0.836 | 0.614 | 0.861 |
| Job crafting | 0.814 | 0.596 | 0.889 |

POQ: perceived overqualification; AVE: average variance extracted; CR: composite reliability.

Next, using the correlation matrix in Table 2, this study compared the correlation coefficients between factors and the squared values of AVE to verify discriminant validity [93]. As the squared values of AVE were higher than the correlation coefficients in all cases, discriminant validity was secured. Based on these results, it was found that there were no issues with the validity and reliability of the measured variables.

This study also conducted the single-factor test proposed by Harman [94], checking for common method bias in the sample data. This study conducted a non-rotational principal-factor analysis [95], and found that the variable with the highest eigenvalue had an explanatory variance of 28.73% and as such determined that the sample bias was not subject to the common method bias.

**Table 2.** Descriptive statistics and correlation analysis.

| Variables | 1 | 2 | 3 | 4 | 5 | 6 | 7 | 8 |
|---|---|---|---|---|---|---|---|---|
| 1. Gender | 1.00 | | | | | | | |
| 2. Age | −0.04 | 1.00 | | | | | | |
| 3. Education level | −0.06 | 0.09 | 1.00 | | | | | |
| 4. Company 1 [a] | 0.03 | 0.02 | 0.03 | 1.00 | | | | |
| 5. Company 2 [a] | −0.11 | 0.03 | −0.01 | −0.24 * | 1.00 | | | |
| 6. POQ | 0.03 | −0.01 | 0.06 | 0.10 | −0.11 | (0.81) | | |
| 7. CA | 0.02 | 0.13 * | 0.04 | −0.02 | 0.00 | 0.03 | (0.78) | |
| 8. JC | −0.10 | 0.06 | 0.08 | 0.09 | −0.16 * | 0.09 | 0.37 ** | (0.77) |
| Mean | - | 31.29 | - | - | - | 3.96 | 4.06 | 3.89 |
| Standard Deviation | - | 4.65 | - | - | - | 0.88 | 1.13 | 0.98 |

$n$ = 257. * $p < 0.05$, ** $p < 0.01$; numbers in brackets are squared AVE values. [a] Companies were coded as a series of dummy variables assigned the largest company (M) for reference. POQ: perceived overqualification, CA: career adaptability, JC: job crafting.

## 4. Results

### 4.1. Descriptive Statistics and Correlation Analysis

The mean, standard deviation, and correlation coefficients of the measurement variables are presented in Table 2. The correlation coefficients between the variables ranged from −0.11 to 0.37. Based on the demographic characteristics of the sample, there were more men (61.8%) than women. The educational levels included high school (25.7%), college (65.2%), and master's degree (10.4%). The average age of the respondents was 31.29 years, with a standard deviation of 4.65. The relationship between career adaptability and job crafting had a strong positive correlation ($r = 0.37$, $p < 0.01$). However, the correlation between POQ and job crafting was not significant ($r = 0.09$, $p > 0.05$). While the linear paired-association between POQ and job crafting was not significant, it is necessary to interpret the results in detail using regression analysis while controlling for other variables to accurately understand the nonlinear relationship.

### 4.2. Analysis of Direct Effects of POQ

To verify the hypothesis of the study model, this study conducted a hierarchical regression analysis (see Table 3). As the values of the independent variable and the squared values have high correlations, this study considered the potential of multicollinearity and conducted mean-centering on all independent variables. While Model 1 found that job crafting was higher in P than in M ($β = −0.15$, $p < 0.05$), the influence of other control variables was insignificant.

**Table 3.** Hierarchical regression analyses.

| Variables | Model 1 | Model 2 | Model 3 | Model 4 |
|---|---|---|---|---|
| Gender | −0.03 | −0.07 | −0.11 | −0.09 |
| Age | 0.01 | 0.02 | 0.02 | 0.00 |
| Education level | 0.04 | 0.05 | 0.06 | 0.05 |
| Company 1 [a] | 0.06 | 0.05 | −0.01 | −0.00 |
| Company 2 [a] | −0.15 * | −0.14 * | −0.12 | −0.13 * |
| POQ | | 0.04 | 0.02 | 0.01 |
| POQ$^2$ | | −0.18 ** | −0.13 * | −0.15 * |
| CA | | | 0.31 ** | 0.27 ** |
| POQ × CA | | | | 0.13 * |
| POQ$^2$ × CA | | | | −0.19 ** |
| R$^2$ | 0.035 | 0.076 * | 0.167 ** | 0.210 ** |
| ΔR$^2$ | | 0.041 * | 0.091 ** | 0.041 * |

$n = 257$. * $p < 0.05$, ** $p < 0.01$; standardized coefficients are reported for regression on job crafting. [a] Companies were coded as a series of dummy variables assigned the largest company (M) for reference. POQ: perceived overqualification, CA: career adaptability, JC: job crafting.

To verify hypothesis 1, this study followed the three-step procedure proposed by Lind and Mehlum [96]. First, according to Model 2, the influence of POQ, which was introduced after controlling for the influence of control variables on job crafting, was not significant ($β = 0.04$, $p > 0.05$), and the squared terms of POQ had a significant negative influence on job crafting ($β = −0.18$, $p < 0.05$). This indicates that the relationship between POQ and job crafting has an inverted U-shape. Second, the relationship between POQ and job crafting was positive at first, though eventually growing weaker as POQ increased, as job crafting decreased after the inflection point (POQ = 4.01; see Figure 1). Third, the slope was sufficiently steep at both ends of the data range. The slope at the low end of job crafting was 1.806, which was positive and significant. The slope at the high end of job crafting was −1.194, which was negative and significant. Based on these analysis results, the hypothesis that POQ has a significant nonlinear influence with an inverted U-shape on job crafting was supported.

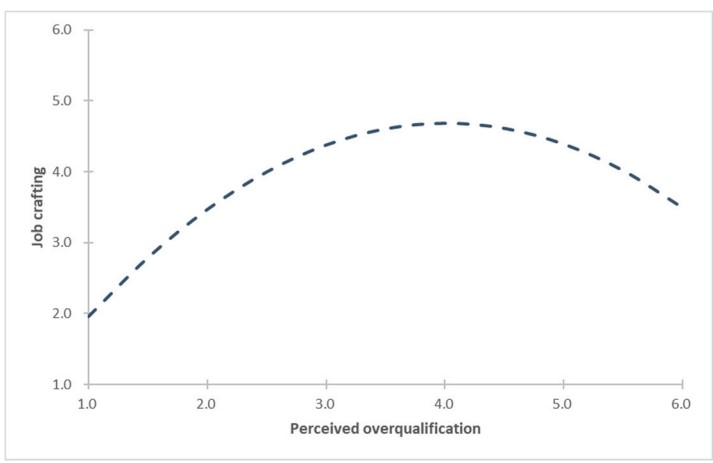

**Figure 1.** Inverted U-shaped relation between POQ and job crafting.

### 4.3. Analysis of Moderating Effects of Career Adaptability

The influence of career adaptability on job crafting was significant in Model 3, as shown in Table 3 ($\beta = 0.31$, $p < 0.01$), and the inverted U-relation of the POQ was also significant ($\beta = -0.13$, $p < 0.05$). This study also confirmed that the interaction variable between the squared term of the POQ and career adaptability had a significant negative relationship with job crafting ($\beta = -0.19$, $p < 0.01$). Based on these results, it is possible to conclude that career adaptability has a moderating effect in the relationship between POQ and job crafting. The $R^2$ of Model 4 was 0.201.

To verify hypothesis 2, additional testing proposed by Haans et al. [97] was conducted. First, as the interaction effect was significant, an interaction graph was drawn using the methods proposed by Aiken and West [98]. A simple slope analysis [99] was also conducted as shown Figure 2. For employees with high career adaptability, that is, within +1 standard deviation from the mean, the nonlinear, inverted U-shaped relationship between POQ and job crafting was very clear ($\beta = -0.33$, $p < 0.01$). The inverted U-shaped nonlinear relationship between POQ and job crafting was also high for members with an average level of career adaptability ($\beta = -0.14$, $p < 0.05$). However, the inflection points of the inverted U-shaped curve were 4.15 and 3.98 for high and average levels of career adaptability, respectively. Furthermore, for employees with low career adaptability, that is, within −1 standard deviation from the mean, the regression coefficient of the second-degree term was meaningless ($\beta = -0.03$, $p > 0.05$).

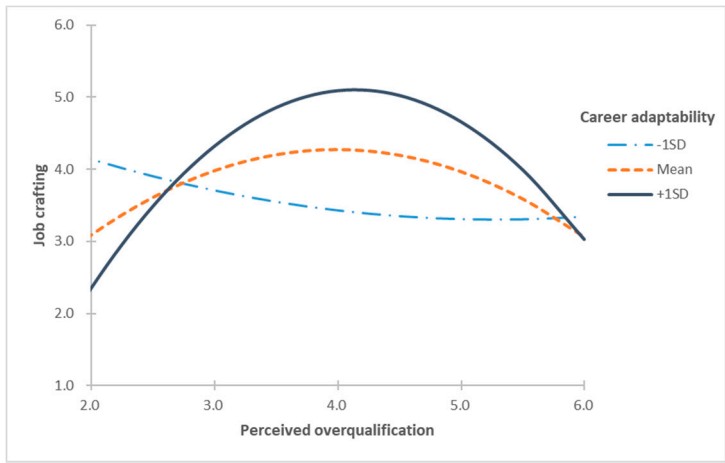

**Figure 2.** Shape flip for the relationship between POQ (*X*-axis) and job crafting (*Y*-axis).

In summary, for members with average or high levels of career adaptability, an appropriate level of POQ has a positive influence on job crafting, while for those with low career adaptability,

POQ did not relate to job crafting. Therefore, hypothesis 2 that stated that career adaptability will have a moderating effect on the relationship between POQ and job crafting was supported. Specifically, higher career adaptability led to strengthening of the inverted U-shaped relationship between POQ and job crafting, and the location of the inflection point rose. However, lower career adaptability was associated with a weaker curved relationship between POQ and job crafting.

## 5. Discussion

With lower economic growth established as the new normal, anxiety about losing employment, underemployment, and unemployment are being intensified. While it is important to shed light on the negative effects of these societal risk factors, it is also important to develop research methods to manage and reduce these factors and their impact. While the qualifications and educational levels possessed by jobseekers and required by employees are on the rise, the issue of overqualification has still emerged, plateauing or lowering the quality of employment. Furthermore, POQ leads to negative perceptions while engaging in simple and dull work that does not match one's skill sets, thus causing employees and firms alike to experience a higher psychosocial toll and lower productivity.

The purpose of this study was to find methods that could lead to the reduction of the negative influences of overqualification and explore its positive effects. For this purpose, this study empirically examined the nonlinear relationship between POQ and job crafting, and the moderating effect of career adaptability. This moderated, quadratic relationship between POQ, job crafting, and career adaptability is the first of its kind in this field of study. The empirical results of this study and their significance can be summarized as follows. First, POQ had a reverse U-shaped nonlinear influence on job crafting. This study found that a POQ value of 4.01 or lower increased job crafting, and beyond this inflection point, the positive influence on job crafting was reduced, showing a curved relationship. This indicates that POQ, if not at extreme levels, can have a positive influence on organizational performance.

Secondly, this study verified that the potential for POQ to induce job crafting differed depending on level of career adaptability. This study analyzed the inverted U-shape according to the recommendations by Lind and Mehlum [96] and Haans, Pieters, and He [97]. Higher career adaptability led to clearer nonlinear relationships between POQ and job crafting. Given high career adaptability, a certain level of POQ increased job crafting. In detail, POQ values of lower than 4.0, i.e., the inflection point, increase job crafting. In other words, unless POQ levels are significant and higher than 4.0, career adaptability strengthened the employees' efforts to overcome POQ through job crafting. However, a lower inflection point was evident with lower levels of career adaptability, and at this point POQ did not act as the driving force of job crafting.

Based on these results, this study presents the following implications and considerations regarding resolving the negative influence of overqualification. First, though past research on POQ has focused on the negative influences of POQ; there is a general lack of research and understanding of the latent positive influences of POQ [21,41]. The results of this study are encouraging as they indicate that appropriate levels of POQ can activate job crafting and that these positive effects can be reinforced by career adaptability. These results can form the foundation for future research on the latent positive influence of POQ and help explore conditions under which employees with POQ can fully utilize their abilities.

Second, this study combined career adaptability and job crafting, which were independently studied in different fields, into a single research model. Every individual has to craft their job to optimize their person–job fit over time [57,64]. This study considered current job design as well as long-term career development. Career adaptability and job crafting are key subjects in the fields of career development and task design and have been recognized as positive factors in overcoming uncertain and rapidly changing work environments. However, there are only a few studies that have combined the two variables, and no research exists on verifying them as variables that can reduce the negative influence of POQ. Therefore, the multidisciplinary approach and results of this study can provide the foundations of new perspectives and research directions for the relevant fields.

Third, this study capitalized career adaptability as a key competency, providing practical implications on negating the side effects of overqualification such as counterproductive behavior [38], job boredom [44], and turnover intention [45] and forming a sound organizational culture. Through this, intervention strategies to reduce underemployment can be developed from the human resources development perspective. New solutions from the perspectives of career development and task design can be derived, applying the results to selection and placement, education and training, and coaching and mentoring. For example, a diagnostic tool that includes preconditions relating to career adaptability and job crafting can be designed as a new and effective standard for hiring and placing employees in boundaryless roles in organizations.

Fourth, the results of this study propose active problem-solving methods for members of organizations who face overqualification. This study demonstrates that individuals can regard POQ as a challenging situation that requires constructive activities and thus attain performance improvements. As such, they are able to improve person–job adaptability through efforts to redefine the traits of their work and recognize its meaning. Furthermore, it is necessary for individuals to move beyond being buried in the present-day overqualification, proactively designing future careers and effectively responding to a challenging reality. As such, firms should provide support for management mechanisms, education, and training that can help improve career adaptability.

Despite the above results and implications, this study has the following limitations. First, this study was unable to control for organizational and team-level variables that can influence the study model, outside of career adaptability at the individual level. To secure a practical explaining power for the results of this study, it is important to consider situational factors such as autonomy at organizational and team levels [3], as well as the POQ levels of colleagues [63]. Second, as POQ and job crafting are related, in that, they continuously have an influence on each other rather than it being a temporary result, there are limitations in discovering a clear causal relationship within the scope of a one-time, cross-sectional study. Future studies should involve longitudinal data collection. Third, this study collected POQ, career adaptability, and job crafting data through a survey, relying on a single response source. While this study has been checked for the potential of bias in the analysis results with the common method using Harman's one-factor method [94], future studies should collect data from various sources to improve internal consistency. Fourth, this study focused on job design and career development and explored the path to alleviate POQ in job crafting and career adaptability. However, the HRM (human resource management) perspective to select and place the right person into the right job was overlooked. Future studies need to consider the effects of HRM policies and practices on realizing Taylor's second principle, "to match workers to their jobs based on capability and motivation and train them to work at maximum efficiency." Lastly, only three companies were surveyed, making it difficult to generalize the results of this study. As such, it is necessary to expand the research participant pool to include multiple industries and firms.

## 6. Conclusions

Studies on perceived overqualification (POQ) have focused on its negative impacts; however, this research shows a reverse U-shaped nonlinear influence of POQ on job crafting and the moderating effect of career adaptability on this relationship. The results of this study indicate that an appropriate level of POQ activates job crafting and that this positive effect is reinforced by career adaptability. Therefore, career adaptability and job crafting can be suggested as practical factors, useful to strengthen an employee's successful career development against the side effects of overqualification.

**Funding:** This research was supported by the National Research Foundation of Korea grant, funded by the Korean Government (NRF-2017S1A5A8022027).

**Acknowledgments:** The author is grateful to SM Consulting Group for data collection.

**Conflicts of Interest:** The author declares no conflict of interest.

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
