# Peer review of "Perceived Overqualification and Job Crafting: The Curvilinear Moderation of Career Adaptability"

_sustainability, doi:10.3390/su122410458_

Round 1
Reviewer 1 Report
I enjoyed reading this paper on perceived overqualification (POQ), job crafting, and the moderating role of career adaptability. The arguments are novel, the findings are interesting, and the insight that the potential negative consequences of POQ can be overcome by job crafting are scientifically as well as practically relevant. I have a number of suggestions to further bolster the strengths of the paper.
MAIN
- Most importantly, the theoretical arguments for the hypotheses currently fall short. Whereas the main concepts are extensively defined and explained, there is very little theory on why the concepts are expected to affect each other. For Hypothesis 1, general references are being made to that "it is commonly known that negative factors such as stress have an inverted U-shaped relation with various performance indicators", but such an argument does not suffice, first of all because stress is not the main focus, and second because this study may present an exception. In fact, there are good reasons to expect a linear positive relationship, because based on the human capital advantage (cf. van Dijk, Shantz, & Alfes, 2020) it can be expected that higher levels of POQ enable individuals to engage more in job crafting. I can imagine that from a certain level of POQ onwards the negative effects based on equity theory (Adams, 1965) and relative deprivation theory (Crosby, 1976) that are generally referred to in the overqualification literature kick in and therefore create an inverted-U shape relationship, but that should be explicitly discussed and specified.
The moderating effect of career adaptability in a similar vein is underspecified. The paragraph from sentences 254 - 264 are difficult to understand, so it's only the next paragraph that provides some clear theoretical argumentation about what kind of a moderating effect is expected and why. Given that it's a moderating effect of a curvilinear relationship, which can occur in many ways, it requires a much more extensive theoretical explanation. Interestingly, when I started thinking about it myself, I already thought that rather than simply strengthening the main relationship, it may actually change it somewhat, given that at low levels of career adaptability it is unlikely that higher levels of POQ result in more job crafting, given that low levels of career adaptability suggest that job crafting anyway as something that is possible and/or of interest. And apparently that is also what you find. Of course, you should not change the hypothesis, but more theoretical explanation of why you argue what you argue is needed; and I think that in the discussion you can then subsequently more elaborately provide a theoretical argument for why you found what you found based on the argument mentioned above.
OTHER
2. In terms of English, grammar, and sentence construction, there was on average 1-2 times per page something that was unclear (e.g., line 30 sentence construction is inaccurate, line 35 Europe is not a country, line 51 focused instead of focus, line 85 improper is not a suitable term, line 281 should be inverted U-shaped relation, so having a professional English copy-editor look at the paper would be really helpful.
3. Line 43-44: the definition doesn't clearly indicate that it's about the overqualified employee's own perception.
4. Line 131-133: It would be helpful to provide more theoretical explanation regarding why and when POQ can lead to more positive outcomes. This relates to the lack of theoretical explanations throughout the paper as indicated in the first comment.
5. Line 134-137: Interesting. You can bolster your argument here by more explicitly indicating that differences may be due to different coping styles.
6. Line 153: Unclear what the 2 dominant claims are that you are referring to. You don't explain them clearly in the next paragraphs.
7. Using a 6-point Likert scale is highly unusual. The reason why an uneven number is common is to give the option of neutral, which with an even number can't be done. So one reference to support this choice isn't enough, please provide more extensive arguments for why a 6-point Likert scale was used.
8. There were three companies so why were only company 1 and 2 included in the analyses? In case company 3 was the base category, then why was that company the base category and not any of the other companies or the combination of the other companies?
9. Line 390-391: Instead of weakening job crafting, shouldn't that be that it didn't relate to job crafting? From what I understood the relationship was nonsignificant?
10. Throughout the paper: At several places it is mentioned that there is a lack of research on the positive consequences of POQ, but there is recently some more attention to potential positive consequences (e.g., Lee, Erdogan, Tian, Willis, & Cao, 2020; van Dijk et al., 2020), so those statements are too strong.
11. The discussion can benefit from some more references throughout.
12. I think the fifth contribution mentioned in the introduction should be removed, because it's not a focus of the paper, or a real test. So at best it's an example.
References
Adams, S. J. (1965). Inequity in social exchange. In L. Berkowitz (Ed.). Advances in experimental social psychology (pp. 267–299). San Diego, CA: Academic Press. Crosby, F. (1976). A model of egoistical relative deprivation. Psychological Review, 83, 85–113. Lee, A., Erdogan, B., Tian, A., Willis, S., & Cao, J. (2020). Perceived overqualification and task performance: Reconciling two opposing pathways. Journal of Occupational and Organizational Psychology. van Dijk, H., Shantz, A., & Alfes, K. (2020). Welcome to the bright side: Why, how, and when overqualification enhances performance. Human Resource Management Review, 30(2), 100688.Author Response
I appreciate for your careful and detail review.
Comments and Suggestions for Authors
I enjoyed reading this paper on perceived overqualification (POQ), job crafting, and the moderating
role of career adaptability. The arguments are novel, the findings are interesting, and the insight that
the potential negative consequences of POQ can be overcome by job crafting are scientifically as well
as practically relevant. I have a number of suggestions to further bolster the strengths of the paper.
I appreciate for your careful and kind review. I could find and modify a number of errors based on your
suggestions.
MAIN
1. (1) Most importantly, the theoretical arguments for the hypotheses currently fall short. Whereas the
main concepts are extensively defined and explained, there is very little theory on why the concepts are
expected to affect each other. For Hypothesis 1, general references are being made to that "it is
commonly known that negative factors such as stress have an inverted U-shaped relation with various
performance indicators", but such an argument does not suffice, first of all because stress is not the
main focus, and second because this study may present an exception…
Thank you for your kind and meaningful comments. In response to your opinion, I have supplemented
Line 112-145 at the end of the section 2.1 and Line 192-220 at section 2.2.
(2) The moderating effect of career adaptability in a similar vein is underspecified. The paragraph from
sentences 254 - 264 are difficult to understand, so it's only the next paragraph that provides some clear
theoretical argumentation about what kind of a moderating effect is expected and why. Given that it's a
moderating effect of a curvilinear relationship, which can occur in many ways, it requires a much more
extensive theoretical explanation…
I have supplemented Line 281-307 to provide more theoretical explanation.
OTHER
2. In terms of English, grammar, and sentence construction, there was on average 1-2 times per page
something that was unclear (e.g., line 30 sentence construction is inaccurate, line 35 Europe is not a
country, line 51 focused instead of focus, line 85 improper is not a suitable term, line 281 should be
inverted U-shaped relation, so having a professional English copy-editor look at the paper would be
really helpful.
As you advised, it was re-checked throughout by a native speaker, including what you pointed out.
3. Line 43-44: the definition doesn't clearly indicate that it's about the overqualified employee's own
perception.
I redefine the concept of POQ.
4. Line 131-133: It would be helpful to provide more theoretical explanation regarding why and when
POQ can lead to more positive outcomes. This relates to the lack of theoretical explanations
throughout the paper as indicated in the first comment.
5. Line 134-137: Interesting. You can bolster your argument here by more explicitly indicating that
differences may be due to different coping styles.
On your 4th and 5th comments, I have present more explanation throughout by reflecting the 10th
comment below. I make an overall complement to this, focusing on hypothesizing the inverted U-shaped
relation in a mixed situation where the positive and negative impacts are simultaneously asserted.
6. Line 153: Unclear what the 2 dominant claims are that you are referring to. You don't explain them
clearly in the next paragraphs.
The sentence structure has been modified to make it easier to understand what you are talking about.
In the beginning, it was intended to illustrate Wrzesniewski & Dutton(2001) and Tims, et al.(2015) by
way of example. As the explanation lengthened, I think, it became difficult to grasp it.
7. Using a 6-point Likert scale is highly unusual. The reason why an uneven number is common is to
give the option of neutral, which with an even number can't be done. So one reference to support this
choice isn't enough, please provide more extensive arguments for why a 6-point Likert scale was
used.
I hope you understand that we simply relied on Chomeya(2010) in our research and used it. As you
indicated, the explanation was insufficient on this point. At the beginning of this study, I found that most
of articles using 4-, 5-, 6-, and 7-point Likert scale did not reveal the reason. I managed to explain it and
employed Chomeya(2010) that 6-point scale can ensure a high level of reliability and discrimination and
avoid having respondents from being neutral by using an even number. I have added further advantage
insisted by Leung(2011). According to Leung, the 6-point scale also have the advantage to follow
normal distributions from Kolmogorov–Smirnov and Shapiro–Wilk statistics.
8. There were three companies so why were only company 1 and 2 included in the analyses? In case
company 3 was the base category, then why was that company the base category and not any of the
other companies or the combination of the other companies?
As you pointed out, it was the dummy variables and company 3 was the reference variable. I have
added this explanation to the table description. When I selected company 3 as reference variable, there
was a general rule to choose the category with a large number of observations. I don't think it really
matters which one I choose as reference variable for this study. Please let me know if you have any
good rule.
9. Line 390-391: Instead of weakening job crafting, shouldn't that be that it didn't relate to job crafting?
From what I understood the relationship was nonsignificant?
I am able to find the incorrect expression. I have modified it according to your comment.
10. Throughout the paper: At several places it is mentioned that there is a lack of research on the
positive consequences of POQ, but there is recently some more attention to potential positive
consequences (e.g., Lee, Erdogan, Tian, Willis, & Cao, 2020; van Dijk et al., 2020), so those
statements are too strong.
I have present some more the positive effects of POQ, especially Line 125-145.
11. The discussion can benefit from some more references throughout.
Following your suggestion, the insufficient references were supplemented in the Discussion section
12. I think the fifth contribution mentioned in the introduction should be removed, because it's not a
focus of the paper, or a real test. So at best it's an example.
I remove the 5th contribution in the Discussion section according to your comment.

Reviewer 2 Report
There is a major difference between the career concept and the job concept. Simply put, the job is something you do to get paid at the end of the month; therefore, it is only a paid job, while a career is a fruitful, rewarding activity. Career is a road, it marks our passage through life, but it is especially related to the activity we carry out and to employment. We have to take an active part in our career, we have to plan it, manage it, follow it, and not wait for the end of the month's salary. Careers can be dynamic, sometimes we move fast, sometimes slow, sometimes we stay longer at a job, and sometimes we change jobs often. Already the extended notion of career becomes clearer, it encompasses all the work tasks I have performed throughout my life. Work is a narrower notion than career and includes all the tasks set for a particular job in the company.
It is possible for a job to turn into a career, but it requires a lot of performance in that field and a continuous effort in terms of personal development. However, if you go into the routine and no longer offer high performance, you risk your own opportunity for development and advancement.
The researched theme has its origins in Taylor's second principle:
Rather than simply assign workers to just any job, match workers to their jobs based on capability and motivation, and train them to work at maximum efficiency.
Selecting the right people for the job is another important part of workplace efficiency.
Considering the specifics of the activity, in telecommunications, the research specifies only the level of training, not the field of specialization, and the level of training is irrelevant in context. When you work in telecommunications it is one thing to have higher education and another to have higher education in telecommunications.
Author Response
I appreciate for your careful and kind review.
Reviewer 2
Comments and Suggestions for Authors
There is a major difference between the career concept and the job concept. Simply put, the job is
something you do to get paid at the end of the month; therefore, it is only a paid job, while a career is
a fruitful, rewarding activity. Career is a road, it marks our passage through life, but it is especially
related to the activity we carry out and to employment. We have to take an active part in our career,
we have to plan it, manage it, follow it, and not wait for the end of the month's salary. Careers can be
dynamic, sometimes we move fast, sometimes slow, sometimes we stay longer at a job, and
sometimes we change jobs often. Already the extended notion of career becomes clearer, it
encompasses all the work tasks I have performed throughout my life. Work is a narrower notion than
career and includes all the tasks set for a particular job in the company.
It is possible for a job to turn into a career, but it requires a lot of performance in that field and a
continuous effort in terms of personal development. However, if you go into the routine and no longer
offer high performance, you risk your own opportunity for development and advancement.
The researched theme has its origins in Taylor's second principle:
Rather than simply assign workers to just any job, match workers to their jobs based on capability and
motivation, and train them to work at maximum efficiency.
Selecting the right people for the job is another important part of workplace efficiency.
Considering the specifics of the activity, in telecommunications, the research specifies only the level
of training, not the field of specialization, and the level of training is irrelevant in context. When you
work in telecommunications it is one thing to have higher education and another to have higher
education in telecommunications.
Thank you for your careful and kind review. I could find and modify a number of shortcomings based
on your suggestions.
I have supplemented, as you give a comment to us, a major difference between the career concept and
the job concept. For example, I have added the sentences like “those who craft their jobs optimize their
person–job fit over time. It is needed to consider not only current short-term job but also future longterm career. We suggest an integrative theoretical framework that draws upon both job design view and
career development view in order to delineate the nature of the relations among POQ, job crafting, and
career adaptability.”
I suggest the specific task of participants for survey, “The survey was conducted with all employees
working in the call centers of the three companies, who had work experience of five years or less. Their
responsibility is relatively simple and normalized, responding to inbound calls from the customers.”
The fourth limitation in Discussion section describes “the HRM perspective to select and place the right
person into the right job was overlooked” and suggests “considering the effects of HRM policies and
systems on realizing Taylor’s second principle” for future researchers.
I appreciate again for your in-depth comments.
This manuscript is a resubmission of an earlier submission. The following is a list of the peer review reports and author responses from that submission.